# Understanding Type 2 Diabetes Mellitus Risk Parameters through Intermittent Fasting: A Machine Learning Approach

**DOI:** 10.3390/nu15183926

**Published:** 2023-09-10

**Authors:** Shula Shazman

**Affiliations:** 1Department of Information Systems, The Max Stern Yezreel Valley College, Yezreel Valley 1930600, Israel; shulas@yvc.ac.il or shulash@openu.ac.il; Tel.: +972-54-6388131; 2Department of Mathematics and Computer Science, The Open University of Israel, Ra’anana 4353701, Israel

**Keywords:** type 2 diabetes mellitus (T2DM), intermittent fasting, machine learning

## Abstract

Type 2 diabetes mellitus (T2DM) is a chronic metabolic disorder characterized by elevated blood glucose levels. Despite the availability of pharmacological treatments, dietary plans, and exercise regimens, T2DM remains a significant global cause of mortality. As a result, there is an increasing interest in exploring lifestyle interventions, such as intermittent fasting (IF). This study aims to identify underlying patterns and principles for effectively improving T2DM risk parameters through IF. By analyzing data from multiple randomized clinical trials investigating various IF interventions in humans, a machine learning algorithm was employed to develop a personalized recommendation system. This system offers guidance tailored to pre-diabetic and diabetic individuals, suggesting the most suitable IF interventions to improve T2DM risk parameters. With a success rate of 95%, this recommendation system provides highly individualized advice, optimizing the benefits of IF for diverse population subgroups. The outcomes of this study lead us to conclude that weight is a crucial feature for females, while age plays a determining role for males in reducing glucose levels in blood. By revealing patterns in diabetes risk parameters among individuals, this study not only offers practical guidance but also sheds light on the underlying mechanisms of T2DM, contributing to a deeper understanding of this complex metabolic disorder.

## 1. Introduction

Type 2 diabetes mellitus (T2DM) is a chronic metabolic disorder characterized by high levels of glucose in the blood due to the body’s inability to effectively use insulin. The incidence of T2DM has been steadily increasing, with estimates suggesting that over 422 million people worldwide currently live with this condition [1,2]. The increasing prevalence of T2DM is largely attributed to changes in lifestyle factors [3,4], such as physical inactivity [5], unhealthy diet [6], and obesity [7].

Although conventional treatments, such as clinical and pharmacological interventions [8,9,10,11], dietary management [6], and exercise plans [5,12,13], for T2DM are available [14,15], they are often associated with high self-discipline, unwanted side effects, and can be expensive.

Therefore, there is a growing interest in exploring lifestyle interventions [4,16,17], such as intermittent fasting (IF), as an effective approach in managing T2DM risk parameters [18,19,20,21,22,23,24,25,26,27]. IF involves periods of reduced or no caloric intake alternated with periods of normal or increased caloric intake. Several studies have suggested that IF may improve various risk parameters associated with T2DM, including insulin sensitivity, glucose metabolism, and inflammation [18,25,26]. Alternate-day intermittent fasting (ADF), for example, demonstrates improvements in diabetes and preservation of beta cell function in polygenic mouse models of T2DM [25,28,29]. In addition, ADF reported as improving endothelial function in T2DM mice [28]. Early time-restricted feeding (eTRF) is an intermittent fasting strategy restricting caloric intake to the first 6–8 h of the day. Furthermore, previous studies in humans have shown that eTRF improves glucose control in adults with prediabetes and high BMI [30].

Intermittent fasting offers a multifaceted advantage in comparison to other interventions for improving T2DM. Unlike sports, which primarily target physical fitness and may require rigorous activity, intermittent fasting offers metabolic benefits through controlled eating patterns. In contrast to strict diets, intermittent fasting allows for more flexible meal timings and offers a sustainable approach that aligns with individuals’ lifestyles. Furthermore, compared to medication regimens, intermittent fasting offers a natural and holistic approach without potential side effects. By optimizing insulin sensitivity, glucose metabolism, and inflammation, intermittent fasting emerges as a versatile and personalized method [31,32,33] that addresses various factors contributing to T2DM.

While most randomized clinical trials studying IF interventions in humans have shown a reduction in T2DM risk parameters, such as fasting glucose and insulin levels, it is important to acknowledge that there are exceptions to this trend. Some studies may not observe significant improvements in these biomarkers following intermittent fasting interventions [18,30]. The reasons for these exceptions could be multifactorial and may include variations in study design, differences in participant characteristics [34], effects of fasting on reproductive hormone levels in humans, especially for menopausal women [35], or variations in the duration or intensity of the intermittent fasting protocols employed. Therefore, before advising IF intervention to individuals with prediabetes or diabetes, it is important to consider the metabolic status of participants prior to the studies, as it can influence the outcomes and interpretation of the findings. The metabolic status of individuals with T2DM can vary widely, including factors such as the duration and severity of diabetes, baseline insulin resistance, level of glycemic control, age, weight, and BMI [19,24]. Tailoring intermittent fasting approaches to everyone’s unique metabolic profile is key to unlocking its full potential in managing T2DM effectively [31,32,36,37,38,39].

This study aims to uncover the underlying patterns and principles that contribute to the successful improvement of T2DM risk parameters using IF. The investigation is based on outcomes of diverse randomized clinical trials implementing various IF interventions in human subjects. Additionally, the study proposes a personalized medicine recommendation system utilizing machine learning algorithms. The primary goal of this personalized medicine recommendation system is to suggest the most effective IF approach for individuals with prediabetes or diabetes, thereby reducing their T2DM risk parameters. The recommendation system considers the impact of gender, age, weight, and BMI on the effectiveness of intermittent fasting in improving T2DM risk parameters.

## 2. Materials and Methods

### 2.1. Intermittent Fasting Interventions

The data for this study were gathered from seven published papers, shown in Table 1, that performed random clinical trials to investigate the effects of IF on T2DM parameters.

The collected data included 838 individuals and 13 different types of interventions. The following interventions, i.e., CER, high carbohydrate, high monounsaturated, DR70, and CCR, included restricted calorie diets or diets with specific food compounds but did not include fasting at all, since they were used for control in the published studies of the random clinical trials. Those interventions were included in the data of this study since they provided interesting information to analyze. For example, in cases where none of the fasting methods improved the T2DM risk parameters of an individual while a calorie restriction did improve them. The fasting interventions based on weekly days that were included in this study were intermittent energy restriction, two days a week trial and not eating on the other five days; fasting every second day; eating only four days a week, intermittent energy and carbohydrate restriction; eating restricted calories only two days a week, intermittent energy and carbohydrate restriction plus free protein and fat; eating restricted calories only two days a week and fasting three non-consecutive days per week; fasting three non-consecutive days per week and on eating days consume 70% energy. The daily morning fasting is a fasting intervention based on day’s hours. In total, records of 838 individuals were collected from 13 different intermittent fasting interventions.

### 2.2. Preparing and Pre-Processing the Data

#### 2.2.1. Selecting the Features

Several features for each of the 838 individuals were collected from the different random clinical trials included in this study. The baseline characteristics contained age, gender, weight, BMI, fasting glucose before and after intervention, and fasting insulin levels before and after intervention. However, the fasting glucose and insulin after intervention have been removed to enable the machine learning classifier to learn the data and predict the results without revealing whether the intervention was successful in terms of reducing T2DM risk parameters.

#### 2.2.2. Selecting Individuals

The motivation of this study is to select the best intervention to reduce T2DM risk parameters. Therefore, only individuals that are candidates for T2DM, pre-T2DM, or have T2DM were considered. Finally, out 838 individuals, 387 were selected with basal glucose above 5 mmol/L or BMI (Body Mass Index) above 25. The baseline characteristics of these 387 individuals are similar to the baseline characteristics of the 838 individuals and contained age, gender, weight, BMI, fasting glucose before and after intervention, and fasting insulin levels before and after intervention. The distribution of the 387 individuals among the 13 different interventions are shown in Figure 1.

#### 2.2.3. Calculating HOMA-IR

T2DM is generally characterized by insulin resistance, where the body does not fully respond to insulin. HOMA-IR stands for Homeostatic Model Assessment of Insulin Resistance. HOMA-IR is calculated using fasting insulin (mU/L) multiplied by fasting glucose (mg/dL). Using HOMA-IR equation, insulin resistance can be estimated from fasting glucose and insulin levels (HOMA-IR = fasting insulin × fasting glucose). A high score of HOMA-IR indicates a significant insulin resistance, which is usually found in people with T2DM. For each individual, we calculated the HOMA-IR difference, which represents the reduction in insulin resistance. The calculation of HOMA-IR difference involves two steps: first, the calculation of HOMA-IR using basal values of fasting glucose and insulin, and second, the calculation of HOMA-IR using values after the intervention or treatment. The HOMA-IR difference is considered True if HOMA-IR before the intervention is higher than HOMA-IR after the intervention. Otherwise, HOMA-IR difference is considered False.

### 2.3. Constructing the Datasets

#### 2.3.1. Dataset to Predict Whether a Specific Intervention Can Improve HOMA-IR

The dataset contained records of 387 individuals from different previous random clinical trials. The aim of this study is to determine whether any of the IF approaches found within the data already published (from random clinical trials) can aid individuals in improving their T2DM risk parameters. Additionally, if multiple interventions demonstrate the potential to enhance T2DM risk parameters for an individual, this study aims to identify the most effective intervention. The interventions included in this study are sourced from various previous random clinical trials. However, each intervention is regarded not just as a singular treatment, but rather as a complete therapy that we would recommend for patients. Consequently, when recommending any of the interventions from this study, we do so along with its entire protocol as outlined in the original clinical study. This protocol encompasses factors such as durations, instructions, diet recommendations, and more.

The dataset contained records of 387 individuals. Each record included properties such as age, weight, BMI, fasting glucose before intervention, fasting insulin levels before intervention, and the type of intervention (1 out of the 13 interventions). The class column contained either True or False, where True means that the HOMA-IR before intervention was higher than the HOMA-IR after intervention. A smaller dataset of 281 derived from the original 387 records dataset. In the smaller dataset, 106 records of control interventions (CER, DR70, and CCR) were removed. In the control intervention (CER, DR70, and CCR), no fasting intervention was applied on the participants.

#### 2.3.2. Continuous Target Column: Improving Fasting Glucose

As described in 4.1, the class for each individual and specific intervention in our dataset was True if the intervention improved HOMA-IR; otherwise, it was False. However, in case we want to compare different interventions, a True or False target column is not useful. Therefore, comparing between different interventions should be performed using target column that represents the difference between HOMA-IR values before and after the intervention. A larger difference indicates a more effective intervention. In that case, the target column in our dataset is continuous instead of binary (True or False). We chose the random forest classifier to predict the difference, since random forest is capable of dealing with continuous values in the target column.

#### 2.3.3. Excluding the Interventions’ Feature

To determine the percentage of the population for which our dataset can recommend an effective intervention to reduce HOMA-IR, we created a new dataset. The dataset was based on the original dataset with target column as True or False, while excluding the “intervention” column.

#### 2.3.4. Increasing the Threshold for Improvement in HOMA-IR or Fasting Glucose

To ensure post-intervention improvement in HOMA-IR, we redefined it as a decrease of over 15% from the normal HOMA-IR value (48,038). The normal HOMA-IR value is calculated based on fasting glucose (5.56 mmol/L) and insulin (80 pmol/L). Normal HOMA-IR = 5.56 × 6 × 80 × 18 = 48,038. Cases with HOMA-IR difference ≥7206 are considered successful (True) in our dataset, while others are False.

### 2.4. Machine Learning Classifiers

This study focuses on classification, determining if specific interventions improve HOAM-IR for prediabetic and diabetic individuals. Four classifiers were chosen: decision tree (J48), logistic model tree, random forest, and logistic regression. Decision trees are both simple and interpretable, with J48 designed to prevent overfitting using pruning. Logistic model trees handle complex relationships, and random forests excel with high-dimensional data scenarios. Logistic regression, on the other hand, is straightforward. These classifiers were chosen due to their respective advantages and compatibility with the dataset. Decision trees manage missing values, random forests perform well in high-dimensional setting, J48 mitigates overfitting, logistic model trees handle intricate relationships, and logistic regression is easy to implement.

### 2.5. Testing Approach

The test approach used is 10-fold, which is building the model using 9 out of 10 data segments and testing on the remaining 1 segment. This process is repeated 10 times. To reinforce the results of the 10-fold test, additional results from an alternative test approach are provided in Appendix A. The additional test approach involves splitting 20% of the data as testing dataset. The model is built using 80% of the data, and the prediction results are then provided for the test set, which comprises 20% of the data.

## 3. Results

### 3.1. Predicting Whether a Certain Intervention Can Improve HOMA-IR

The initial step in selecting the optimal intervention for a patient with reported prediabetes is to address the question: Does a specific intervention improve HOMA-IR? Four machine learning classifiers were chosen to answer this question: J48 decision tree, logistic model tree (LMT), random forest, and logistic regression. Detailed information about each classifier can be found in the Method Section 2.4. The dataset described in the Methods section was trained and tested, utilizing the 10-fold test to assess the predictive ability of each classifier for HOMA-IR improvement. Improvement in HOMA-IR was defined as a decrease in HOMA-IR after the intervention compared to before the intervention. The results for each classifier can be found in Table 2 under the “Discrete difference” row and “HOMA-IR (with control)” column. The results show modest significance, with AUC values ranging from 0.6 to 0.7, and accuracy ranging from 70% to 72%. To reduce data noise, 106 records of control interventions (CER, DR70, and CCR) were removed from the dataset. These control interventions did not involve fasting. After removing these records (see Methods section for details), slight improvements in prediction were observed for all four classifiers. The results can be found in Table 2 under the “Discrete difference” row and “HOMA-IR (no control)” column. The significance remained moderate, with AUC values ranging from 0.65 to 0.71 and accuracy ranging from 68% to 74%. Interestingly, when the difference between post-intervention and baseline HOMA-IR increased, the results became more significant in terms of AUC and accuracy. In this case, improvement in HOMA-IR was defined as a decrease in HOMA-IR after the intervention by more than 15% compared to the normal HOMA-IR (calculated using fasting glucose of 5.56 mmol/L and fasting insulin of 80 pmol/L; calculation explained in the Methods section). Appendix A present AUC and accuracy results for different fasting glucose and HOMA-IR differences cutoffs (before and after interventions) of 15%, 10%, and 20%, respectively. The Appendix A demonstrate that the 15% cutoff performs the best. The results for each classifier, using the cutoff of 15%, can be found in Table 2 under the “Discrete difference above 15%” row and “HOMA-IR (no control)” column. AUC values ranged from 0.73 to 0.89, and accuracy ranged from 74% to 82%. The logistic model tree classifier exhibited the most favorable results, with an AUC of 0.89 and an accuracy of 82%. These findings suggest that the proposed method can serve as a robust foundation for a recommendation system. Table 2 shows that using the logistic model tree classifier, HOMA-IR improvement can be predicted with an accuracy of 82% and an AUC of 0.89. The other three classifiers also provided similar predictive results.

### 3.2. Can We Predict Improvement in HOMA-IR without Knowing the Intervention?

Establishing a robust foundation for a recommendation system involves assessing the predictability of HOMA-IR improvement and the accuracy of predictions. To determine this, we can leverage our dataset without the intervention feature. Additionally, it would be intriguing to investigate the decision rules for HOMA-IR improvement by training and testing our data without the intervention column. The results of this test, using the four classifiers, are presented in Table 2 under the “Discrete difference above 15%. No Interventions” row and “HOMA-IR (no control)” column. The AUC values range from 0.71 to 0.88, with accuracy ranging from 72% to 83%. While these results are slightly less significant than when utilizing the intervention column (AUC between 0.73 and 0.89 and accuracy between 74% and 82%), this is reasonable as omitting the intervention column leads to some information loss. Nonetheless, these results are meaningful and support the recommendation system. For instance, the logistic model tree classifier demonstrates an accuracy of 83% and an AUC of 0.88, while the random forest yields an accuracy of 79% and an AUC of 0.86. Interestingly, Figure 2 highlights the significance of fasting glucose and fasting insulin as influential factors in predicting HOMA-IR improvement using the J48 classifier. Furthermore, the figure reveals the substantial role of BMI in determining the appropriateness of IF for reducing HOMA-IR in individuals below 59 years of age. In contrast, for individuals aged 59 and above, gender, specifically for women, appears to have a higher likelihood of benefiting from an IF approach to reduce HOMA-IR.

### 3.3. Predicting Whether a Specific Intervention Can Improve Fasting Glucose Only

In real life, patients typically undergo only fasting glucose blood tests as part of their routine health check-ups, while fasting insulin tests are not commonly included in the standard blood tests conducted by Health Maintenance Organizations (HMOs). Therefore, a recommendation system based solely on fasting glucose levels would have lower performance compared to a system based on HOMA-IR. However, it is still intriguing to evaluate the performance of our data when using only fasting glucose.

The results of the glucose test can be found in Table 2 under the “Discrete difference above 15%. No Interventions” row and “Fasting Glucose (control)” column. Interestingly, the accuracy of the prediction ranges from 93% to 95%, compared to 74% to 77% with the same test for HOMA-IR. Additionally, the AUC values show improvement, ranging from 0.64 to 0.91, compared to 0.75 to 0.83 for the HOMA-IR test. The results of the glucose test without control or interventions, as shown in Table 2, are similar to those of HOMA-IR tests but with better accuracy and AUC values.

The higher accuracy when using only fasting glucose can be attributed to the smaller number of successful cases in improving fasting glucose, compared to the number of cases with improved HOMA-IR. However, the improvement in AUC is not affected by these numbers, indicating that the data contains relevant information.

Figure 3 illustrates the decision tree based on the J48 classifier using only fasting glucose without the intervention columns but with control. Six decision pathways are depicted, with four resulting in a False decision and two leading to a True decision. Lower basal fasting glucose tends to lead to a False decision, while being a female with lower weight favors a True decision. Conversely, being a female with higher weight leans towards a False decision. Additionally, older males tend to result in a False decision, whereas younger males tend to result in a True decision. These results are logical and allow us to conclude that weight is the crucial feature for females, while age is the determining factor for males.

### 3.4. Comparison of Different Interventions in Improving T2DM Risk Parameters Using Continuous Difference

The previous sections extensively discussed the ability and success in predicting whether an intervention would improve T2DM risk parameters for a individuals with reported prediabetes or diabetes. The next crucial question in developing a recommendation system is to determine the most effective IF method for an individual. This question can be addressed through continuous difference analysis. In this approach, the target column represents the difference between the baseline fasting glucose level and the fasting glucose level after the intervention. In previous analyses, the target column was labeled as True when the baseline fasting glucose level was higher than the fasting glucose level after intervention; otherwise, it was labeled as False. However, with the continuous column as the target, we can select the method with the highest difference as the best match. By using a continuous column as the target, the solution shifts from binary classification (True or False) to predicting the value of the difference. The success of this prediction can be measured using the correlation coefficient between the actual and predicted values. For this prediction, the random forest algorithm was employed. The results can be found in Table 2 under the “Continuous difference” row and “Fasting Glucose (control)” column, with the random forest algorithm achieving a correlation coefficient of 0.51. The significance of these results will be discussed in the next section, where the correlation coefficients of these algorithms will be compared with a random dataset.

Examples of selecting the optimal IF intervention using the recommendation system can be found in Appendix A, where Appendix A recommends the “diet high mono” intervention, while Appendix A suggests the “CCR” intervention.

### 3.5. Random Testing

An interesting and essential question in prediction analysis is whether the same results can be obtained randomly. To address this, we utilized the same dataset but with a randomized target column, while maintaining the proportion between True and False labels identical to the original target column in each test. The results of all random tests can be found in Appendix A. Notably, all random tests consistently exhibited significantly lower success rates compared to the real tests. For instance, Appendix A presents the results of the dataset with a target column indicating a 15% difference, while Appendix A displays the random results for the same dataset. In the real test, the AUC for fasting glucose without control and with intervention ranged from 0.82 to 0.93, whereas in the random test, the AUC ranged from 0.47 to 0.57. Appendix A demonstrates the random test for continuous difference, where the target column was replaced by random continuous values within the range of the highest and lowest differences found in the original column of the dataset (−2 to 2 mg/dL). As seen in Appendix A, the correlation coefficients for all random tests are significantly lower than the results of the original test. These outcomes emphasize the fact that the successful predictions described in the previous sections cannot be achieved randomly.

## 4. Discussion

Over the past decade, the landscape of T2DM care has witnessed remarkable progress, ushering in a new era of personalized and holistic approaches.

Utilizing cutting-edge methods, treating T2DM has taken innovative paths that hold promising potential. Stem cell therapy, for instance, represents a forward-looking approach that aims to harness the regenerative capabilities of stem cells to address the underlying factors of T2DM [47,48,49]. This method targets the restoration of damaged pancreatic beta cells responsible for insulin production, thereby enhancing the body’s glucose regulation. Stem cells sourced from various origins, such as adipose tissue or bone marrow, offer a way to replenish beta cell populations and mitigate the inflammatory response associated with T2DM. While stem cell therapy is in its early stages, preliminary clinical trials and preclinical studies have shown encouraging results, with improved glycemic control noted in certain patients. However, challenges like selecting optimal cell sources and ensuring long-term efficacy remain [50].

In addition to stem cell therapy, another emerging approach gaining traction is the use of CRISPR-Cas9 gene editing technology to modify genes related to T2DM [51]. This innovative method aims to target the root causes of the disorder by directly manipulating key genes involved in insulin production and glucose regulation. Although still in its infancy, precision gene editing offers the potential for more focused and durable interventions. This strategy directly tackles genetic factors contributing to T2DM and has the potential not only to manage but also to potentially reverse the condition’s progression. However, thorough research and clinical trials are essential to fully understand its safety, efficacy, and long-term impacts [52].

Among the various avenues explored, IF has garnered considerable attention as an alternative approach to conventional T2DM treatments [10,18,19,21,22,53]. IF entails cyclic patterns of controlled eating and fasting, demonstrating the potential to enhance insulin sensitivity, enhance glucose metabolism, and reduce inflammation. Unlike strict diets or exercise regimens, IF is adaptable to individuals’ lifestyles and embraces a natural, holistic methodology without significant side effects.

IF primary mechanisms to improve T2DM risk parameters involve metabolic changes that enhance overall metabolism and trigger tissue-specific metabolic adaptations. These adaptations include modifications in the gut microbiota, remodeling of adipose tissue, restoration of circadian rhythm balance, and increased autophagy in peripheral tissues [19,53].

IF offers a promising approach for treating T2DM, though individual responses can vary based on factors such as age, metabolic profile, and overall health. While some experience significant improvements, others may observe minimal changes, highlighting the need for personalized diabetes management. To address this, a recommendation system powered by machine learning analyzes individual characteristics to tailor IF guidance, maximizing its benefits. Age, weight, and BMI also play crucial roles, influencing outcomes as metabolic conditions differ. Recognizing these factors is vital for understanding conflicting study results and comprehensively evaluating IF’s potential benefits and limitations for T2DM intervention.

IF, however, does not exist in isolation. Rather, it complements and enriches the existing arsenal of treatments. Traditional approaches continue to hold value in managing T2DM, especially when tailored to each patient’s needs. Combining IF with pharmacological interventions and exercise can create a comprehensive regimen that addresses the multifaceted nature of T2DM. Moreover, the synergy of IF with advancements in precision medicine further refines treatment strategies. Utilizing machine learning algorithms to recommend personalized IF plans [54] aligns with the broader trend of precision medicine, where interventions are customized to individuals based on genetic, metabolic, and lifestyle factors.

The results of this study allow us to conclude that weight is the crucial feature for females, while age is the determining factor for males to reduce glucose levels in blood. Furthermore, the results reveal the substantial role of BMI in determining the appropriateness of IF for reducing HOMA-IR in individuals below 59 years old. In contrast, for individuals aged 59 and above, gender, specifically for women, appears to have a higher likelihood of benefiting from the IF approach in reducing HOMA-IR. Leveraging advanced machine learning techniques, such a recommendation system holds the potential to provide highly personalized and customized recommendations, thereby optimizing the advantages of intermittent fasting for various subgroups within the population. Moreover, the development of such a recommendation system will contribute to our understanding of the underlying mechanisms behind T2DM and explore potential clinical applications of intermittent fasting in a more precise and individualized manner.

## 5. Conclusions

Over the last ten years, significant transformations have occurred in the approach and treatment of T2DM. Among these changes, IF has emerged as a holistic and individualized method that complements traditional approaches. IF opens a promising avenue for addressing the intricacies of T2DM, utilizing inherent mechanisms to enhance health outcomes. Through the integration of IF with established methods and precision medicine, the landscape of T2DM management could experience a revolutionary shift, tailoring strategies to each patient for optimized overall well-being. This study introduces a personalized recommendation system aimed at promoting health using diverse IF strategies. Moreover, it delves into the discovery of concealed patterns and fundamental principles that contribute to advancing T2DM management by mitigating risk factors through IF application.

## Figures and Tables

**Figure 1 nutrients-15-03926-f001:**
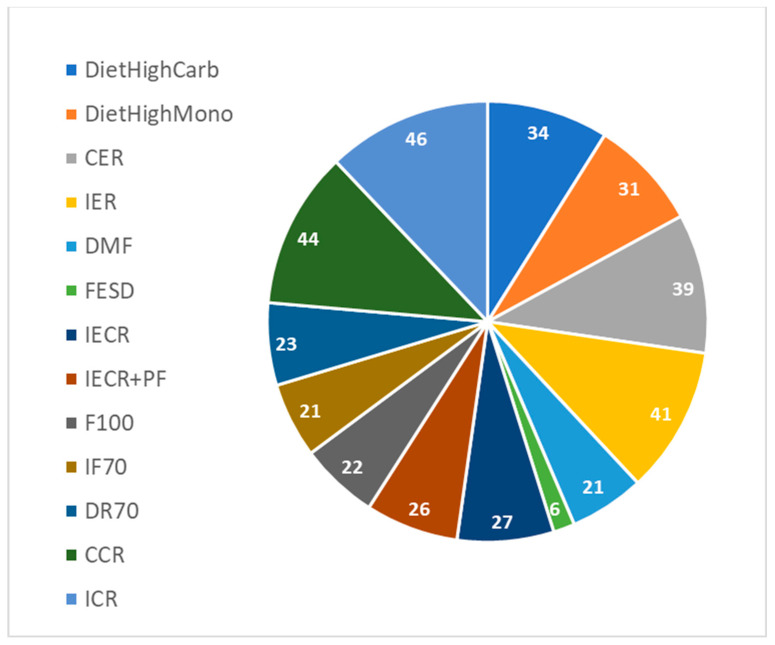
The distribution of the 387 individuals among the 13 different interventions.

**Figure 2 nutrients-15-03926-f002:**
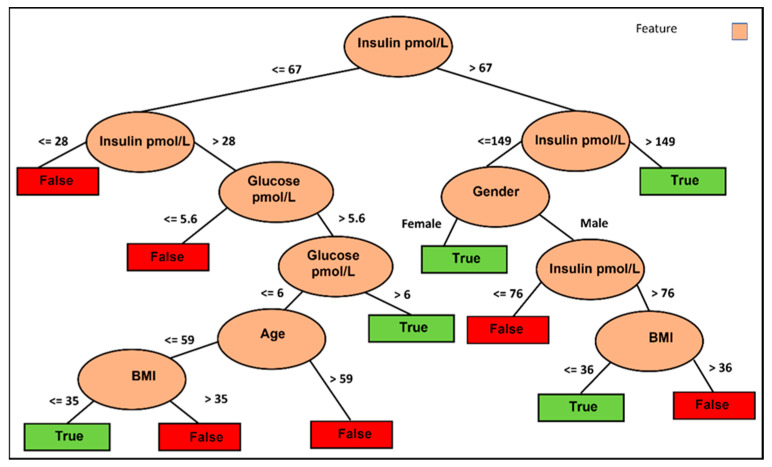
Informative features for selecting the best IF approach to improve T2DM for individuals. Visualization of the J48 decision tree predicting more than 15% improvement in HOMA-IR.

**Figure 3 nutrients-15-03926-f003:**
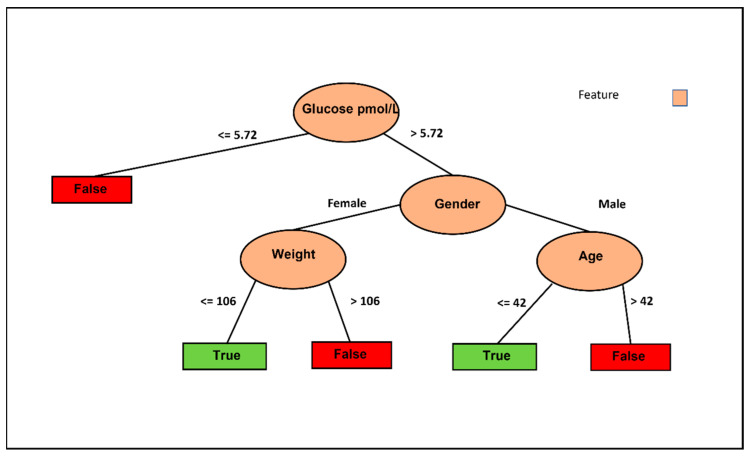
Informative features to select the optimal IF approach to improve T2DM for individuals. Visualization of the J48 decision tree predicting more than 15% improvement in fasting glucose.

**Table 1 nutrients-15-03926-t001:** IF regimens.

Intervention Name	Details	Reference
CER	Continuous energy restriction—7-days-a-week trial; eating restricted calories every day.	Harvie et al., 2011 [40]
IER	Intermittent energy restriction, 2-day-a-week trial; eating restricted calories only two days a week.	Harvie et al., 2011 [40]
DMF	Daily morning fasting; start eating at noon and finish at 20:00.	Chowdhury et al., 2016 [41]
FESD	Fasting every second day; eating only four days a week.	Halberg et al., 2005 [42]
IECR	Intermittent energy and carbohydrate restriction; eating restricted calories only two days a week.	Harvie et al., 2013 [43]
IECR + PF	Intermittent energy and carbohydrate restriction + free protein and fat; eating restricted calories only two days a week.	Harvie et al., 2013 [43]
High Carb	High carbohydrate weight loss diet; eating restricted calories every day.	Clifton et al., 2004 [44]
High Mono	High monounsaturated weight loss diet; eating restricted calories every day.	Clifton et al., 2004 [44]
IF100	Fasting three non-consecutive days per week.	Hutchison et al., 2019 [45]
IF70	Fasting three non-consecutive days per week and on eating days have 70% energy.	Hutchison et al., 2019 [45]
DR70	Seven days a week with 70% energy.	Hutchison et al., 2019 [45]
CCR	Daily energy deficit ∼20%.	Ruth Schübel et al., 2018 [46]
ICR	Fasting two non-consecutive days per week and on eating days have 75% energy.	Ruth Schübel et al., 2018 [46]

**Table 2 nutrients-15-03926-t002:** Area under curve (AUC) (left value) and accuracy (right value) of predicting fasting glucose or HOMA-IR difference.

		Fasting Glucose		HOMA-IR	
		With Control	No Control	With Control	No Control
Discrete difference	J48	0.66 65%	0.67 67%	0.68 70%	0.65 68%
	LMT	0.72 67%	0.73 66%	0.60 72%	0.70 73%
	Random forest	0.71 68%	0.70 65%	0.68 70%	0.71 71%
	Logistic	0.72 68%	0.73 66%	0.70 71%	0.70 74%
Discrete difference. No interventions	J48	0.61 63%	0.63 65%	0.57 68%	0.54 68%
	LMT	0.70 64%	0.72 66%	0.65 70%	0.62 70%
	Random forest	0.68 63%	0.68 64%	0.60 69%	0.63 72%
	Logistic	0.71 65%	0.71 66%	0.65 70%	0.64 71%
Discrete difference above 15%.	J48	0.79 93%	0.82 96%	0.74 74%	0.73 74%
	LMT	0.90 94%	0.91 96%	0.83 75%	0.89 82%
	Random forest	0.90 95%	0.93 96%	0.82 76%	0.87 79%
	Logistic	0.82 95%	0.82 96%	0.83 76%	0.88 82%
Discrete difference above 15%. No interventions	J48	0.64 93%	0.73 94%	0.75 75%	0.74 73%
	LMT	0.91 95%	0.90 95%	0.82 74%	0.82 76%
	Random forest	0.91 95%	0.92 95%	0.82 76%	0.82 75%
	Logistic	0.90 95%	0.94 95%	0.83 77%	0.82 78%
Continuous difference	Random forest	0.51	0.51	0.36	0.46

## Data Availability

The table containing the data of this research may be found at the following link: https://github.com/shulash/IntermittentFasting/blob/main/training%20data%20387.csv (accessed on 5 August 2023). Additional data will be provided upon request shulash@openu.ac.il.

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
