# Peer review of "Understanding Type 2 Diabetes Mellitus Risk Parameters through Intermittent Fasting: A Machine Learning Approach"

_nutrients, 2023, doi:10.3390/nu15183926_

Round 1
Reviewer 1 Report
The authors present a study about prediction of Type 2 Diabetes risk via intermittent fasting. The idea is good, and the structure is well-organized. However, there are a couple of major concerns to be addressed.
The authors included a few published studies to predict the risk of diabetes. What are the baseline characteristics for 411 people, i.e., race, gender. Figure 1 shows the distribution of intervention for 838 people. How about the 411 people. Are they the same?
Since the data was collected from multiple prior clinical studies, how the author integrate the heterogeity for these data?
10-fold cross validation could not serve as the real testing approach. Please split 20% of the data as test dataset.
Decision tree often fail to overfitting issue. Does the author consider it as well. Please provide the result on training and test set.
The format is not optimal. Please check the spells and format carefully.
Author Response
Reviewer 1 comments
Thank you for your valuable comments.
Please see my answers bellow.
The authors present a study about prediction of Type 2 Diabetes risk via intermittent fasting. The idea is good, and the structure is well-organized. However, there are a couple of major concerns to be addressed.
This study proposes a personalized recommendation system to promote health through various intermittent fasting strategies. Furthermore, the study seeks to uncover hidden patterns and underlying principles that contribute to enhancing the management of Type 2 diabetes mellitus (T2DM) by reducing risk parameters through the application of intermittent fasting (IF).
The authors included a few published studies to predict the risk of diabetes. What are the baseline characteristics for 411 people, i.e., race, gender.
In order to be clearer and precise regarding this question the following text was added to the article lines 138-140. "The baseline characteristics of the 411 individuals is similar to the baseline characteristics of the 838 individuals and contained age, gender, weight, BMI, fasting glucose before and after intervention and fasting insulin before and after intervention".
Figure 1 shows the distribution of intervention for 838 people. How about the 411 people. Are they the same?
The control intervention has 24 individuals, therefore, in total only 387 (and not 411) individuals were collected from 13 different IF interventions. Figure1 shows the distributions of the 387 individuals among the 13 different interventions. The location of the figure in the text is changed accordingly. Lines 140-141 were added to better explain the figure: " The distributions of the 387 individuals among the 13 different interventions are shown in Figure 1."
Since the data was collected from multiple prior clinical studies, how the author integrate the heterogeity for these data?
The aim of this study is to determine whether any of the intermittent fasting approaches found within the data already published (from Random Clinical Trials) can aid individuals in improving their Type 2 Diabetes risk parameters. Additionally, if multiple interventions demonstrate the potential to enhance T2DM risk parameters for an individual; this study aims to identify the most effective intervention. The interventions included in this study are sourced from various previous random clinical trials, the common baseline characteristics of the individuals in all clinical studies included in this study are identical (age, gender, weight, BMI, fasting glucose before and after intervention and fasting insulin before and after intervention). However, each intervention is regarded not just as a singular treatment, but rather as a complete therapy that we would recommend for patients. Consequently, when recommending any of the interventions from this study, we do so along with its entire protocol as outlined in the original clinical study. This protocol encompasses factors such as duration, instructions, diet recommendations, and more.
Lines 172-183 were added to the paper's text to further explain this point.
10-fold cross validation could not serve as the real testing approach. Please split 20% of the data as test dataset.
|
HOMA-IR |
Fasting Glucose |
|
|||||
|
No control |
With control |
No control |
With control |
|
|||
|
0.65 0.7 (0.85) 68% 65% (84%) |
0.68 0.6 (0.87) 70% 68% (85%) |
0.67 0.78 (0.83) 67% 67% (69%) |
0.66 0.61 (0.81) 65% 62% (69%) |
J48 |
Discrete difference |
||
|
0.70 0.7 (0.72) 73% 77% (78%) |
0.60 0.6 (0.62) 72% 69% (85%) |
0.73 0.74 (0.78) 66% 64% (68%) |
0.72 0.73 (0.77) 67% 64% (69%) |
LMT |
|||
|
0.71 0.75 (0.87) 71% 70% (93%) |
0.68 0.66 (0.85) 70% 68% (93%) |
0.70 0.74 (0.76) 65% 67% (69%) |
0.71 0.71 (0.82) 68% 66% (73%) |
Random forest |
|||
|
0.70 0.73 (0.74) 74% 76% (76%) |
0.70 0.62 (0.74) 71% 67% (73%) |
0.73 0.88 (0.88) 66% 69% (69%) |
0.72 0.81 (0.88) 68% 66% (71%) |
Logistic |
|||
|
0.54 0.61 (0.64) 68% 71% (78%) |
0.57 0.58 (0.69) 68% 62% (76%) |
0.63 0.82 (0.85) 65% 66% (66%) |
0.61 0.60 (0.61) 63% 62% (66%) |
J48 |
Discrete difference. No interventions |
||
|
0.62 0.62 (0.64) 70% 74% (74%) |
0.65 0.6 (0.73) 70% 69% (80%) |
0.72 0.75 (0.77) 66% 65% (67%) |
0.70 0.71 (0.72) 64% 62% (65%) |
LMT |
|||
|
0.63 0.65 (0.8) 72% 69% (96%) |
0.60 0.61 (0.78) 69% 65% (93%) |
0.68 0.74 (0.76) 64% 64% (69%) |
0.68 0.69 (0.77) 63% 62% (68%) |
Random forest |
|||
|
0.64 0.66 (0.78) 71% 75% (75%) |
0.65 0.59 (0.66) 70% 66% (71%) |
0.71 0.71 (0.72) 66% 65% (67%) |
0.71 0.72 (0.74) 65% 61% (65%) |
Logistic |
|||
|
0.73 0.81 (0.96) 74% 73% (92%) |
0.74 0.65 (0.93) 74% 72% (89%) |
0.82 0.92 (0.97) 96% 96% (98%) |
0.79 0.64 (0.96) 93% 91% (98%) |
J48 |
Discrete difference above 15%. |
||
|
0.89 0.82 (0.82) 82% 81% (81%) |
0.83 0.69 (0.74) 75% 72% (88%) |
0.91 0.93 (0.97) 96% 94% (98%) |
0.90 0.91 (0.94) 94% 92% (97%) |
LMT |
|||
|
0.87 0.88 (1.0) 79% 76% (100%) |
0.82 0.81 (1.0) 76% 75% (100%) |
0.93 0.94 (0.98) 96% 96% (98%) |
0.90 0.89 (1.0) 95% 93% (100%) |
Random forest |
|||
|
0.88 0.91 (0.92) 82% 84% (84%) |
0.83 0.75 (0.87) 76% 72% (78%) |
0.82 0.97 (0.97) 96% 97%(97%) |
0.82 0.90 (0.96) 95% 92% (97%) |
Logistic |
|||
|
0.74 0.8 (0.85) 73% 76% (83%) |
0.75 0.7 (0.87) 75% 69% (83%) |
0.73 0.92 (0.92) 94% 95%(95%) |
0.64 0.61 (0.93) 93% 94% (96%) |
J48 |
Discrete difference above 15%.No interventions |
||
|
0.82 0.82 (0.84) 76% 81% (81%) |
0.82 0.7 (0.9) 74% 74% (85%) |
0.90 0.93 (0.95) 95% 94% (96%) |
0.91 0.92 (0.93) 95% 92% (95%) |
LMT |
|||
|
0.82 0.85 (1.0) 75% 73% (100%) |
0.82 0.77 (1.0) 76% 72% (100%) |
0.92 0.98 (1.0) 95% 95%(100%) |
0.91 0.92 (1.0) 95% 94% (100%) |
Random forest |
|||
|
0.82 0.84 (0.86) 78% 82% (82%) |
0.83 0.72 (0.84) 77% 73% (78%) |
0.94 0.94 (0.95) 95% 94% (96%) |
0.90 0.91 (0.93) 95% 91% (95%) |
Logistic |
|||
|
0.46 |
0.36 |
0.51 |
0.51 |
Random Forest |
Continuous difference |
||
In order to strength the 10-fold test results, an additional test approach results are shown in the upper table which is added as supplementary Table S7. The additional test approach is split 20% of the data as test dataset. Each cell in the table contains two lines. The upper line shows the AUC results while the lower line describes the accuracy results. Every line composed of 3 columns the left column shows the result of 10-fold test, the middle column shows the results of splitting 20% of the data as test dataset test while the right column (in parentheses) contains the training results.
The results found in the table show very similar values with minor changes between both tests 10-fold and split 20%. These results strength the 10-fold test results.
Decision tree often fail to overfitting issue. Does the author consider it as well.
There are 4 classifiers used in this study in order to validate and strength the results. One of the 4 is Logistic regression classifier which is not a decision tree classifier. In addition, another classifier in this study the J48 algorithm, which is an implementation of the C4.5 decision tree algorithm, includes a pruning step to help avoid overfitting. Furthermore, by combining bagging (Bootstrap Aggregating) and feature randomness, Random Forest (a classifier also used in this study) creates a robust and generalizable model that is less prone to overfitting. This point is further discussed in paragraph 2.4.
Please provide the result on training and test set.
The table above (which is added as supplementary Table S7) contains results on training and test set.

Reviewer 2 Report
Dear Authors,
the manuscript appears interesting to me, however, requires further improvement and some revisions.
Here are my comments and recommendation:
Introduction
The introduction may lack a few arguments with respective references to improve the red line of the Introduction and to make it more interesting for a broad spectrum of readers, who are not 100% experts, but who are also interested in this field of treating T2DM including new approaches, particularly such as interval fasting and sport interventions.
In the Introduction, the authors stated that sport or exercise interventions are new and promising approaches to treating T2DM, however, the authors did not adequately cite the essential references of this new field. Please improve the introduction by citing the following refs 1. Joaquim L, et al. J Physiol Biochem 2022. doi:10.1007/s13105-021-00839-4. 2. Grajower MM & Horne BD. Nutrients 2019. doi: 10.3390/nu11040873. 3. Mourouti N et al. Nutrients 2023 doi:10.3390/nu15143155. 4 Marshe VS, et al., Prog Neuropsychopharmacol Biol Psych 2017 doi:10.1016/j.pnpbp.2017.07.026. 5. Wibowo RA et al. Int J Environ Res Public Health 2022. doi:10.3390/ijerph19074199.
Methods
At the end of the method section, to be more precise in Paragraph 2.5, the authors inadequately made a discrimination how to present the results: „3. Results […].“ Authors should arias these sentences because this confuses all readers.
Results
Overall, the result section appears too long and detailed; therefore, the results could perhaps be shortened a bit to underline the main developments. However, this is just a suggestion by the reviewer, if the authors do not agree, it might be ok to just slightly change it, or perhaps leave it in its current form.
Discussion
The discussion is far too superficial. The authors should write a real discussion that integrates the current finding of the study into the bigger picture regarding the broad spectrum of literature of the last 5 to 10 years, for example. The current Discussion is scientifically insufficient! The authors should be guided by the literature (including the cited refs) suggested by the reviewer for the introduction. Despite stell cell medicine being a domain mainly for T1Dm because of its specific pathophysiology; Moreover, other novel stem cell approaches to cure particularly T2Dm should be mentioned and critically discussed. The authors should briefly summarize the main findings of the following papers and cite here: 1.) Zayzafoon M et al., J Cell Biochem 2000. doi: 10.1002/1097-4644(20001101). 2.) Yu S, et al., Stem Cell Res Ther 2019. doi:10.1186/s13287-019-1474-8. 3..) Bentz K et al., Cell Physiol Biochem 2010. doi:10.1159/000323991. 4.) Päth G et al. Metabolism 2019. doi:10.1016/j.metabol.2018.10.005. However, using these approaches requires caution, please discuss and cite: 1.) Molcanyi M, et al., J Neurosci Methods 2013. doi: 10.1016/j.jneumeth.2013.02.012. 2) Bora J et al., Naunyn Schmiedebergs Arch Pharmacol 2023. doi:10.1007/s00210-023-02631-1.
Conclusions
There is a headline titled "5. Conclusions" however any text body is missing?! The author should choose either a conclusive sentence as the final one of the discussion or a full paragraph #5 that draws and summarizes the conclusions of this paper. But in its current form, the manuscript text does not make sense at the end of the discussion. The authors have to revise this part.
Overall, a critical or at best a native speaker should double-check the paper regarding typos and English grammar and spelling one on hand, and perhaps all authors should help to substantially improve the current manuscript version.
Kind regards,
Your reviewer.
Author Response
Reviewer 2 comments
Thank you for your valuable comments.
Please see my answers bellow.
the manuscript appears interesting to me, however, requires further improvement and some revisions.
Here are my comments and recommendation:
Introduction
The introduction may lack a few arguments with respective references to improve the red line of the Introduction and to make it more interesting for a broad spectrum of readers, who are not 100% experts, but who are also interested in this field of treating T2DM including new approaches, particularly such as interval fasting and sport interventions.
The introduction has been enriched (additional references were added) to encompass novel methodologies for treating T2DM, and to provide a comprehensive explanation of the concept of a personalized recommendation system aimed at enhancing Type 2 diabetes risk parameters through the utilization of intermittent fasting suggested by the current study.
In the Introduction, the authors stated that sport or exercise interventions are new and promising approaches to treating T2DM, however, the authors did not adequately cite the essential references of this new field. Please improve the introduction by citing the following refs
- Joaquim L, et al. J Physiol Biochem 2022. doi:10.1007/s13105-021-00839-4.
This reference was already included in the original text as ref. #12, in the revised version it is ref. #19.
- Grajower MM & Horne BD. Nutrients 2019. doi: 10.3390/nu11040873.
This reference is added as ref. #10.
- Mourouti N et al. Nutrients 2023 doi:10.3390/nu15143155.
This reference is added as ref. #3.
4 Marshe VS, et al., Prog Neuropsychopharmacol Biol Psych 2017 doi:10.1016/j.pnpbp.2017.07.026.
This reference is added as ref. #11.
- Wibowo RA et al. Int J Environ Res Public Health 2022. doi:10.3390/ijerph19074199.
This reference is added as ref. #13.
Methods
At the end of the method section, to be more precise in Paragraph 2.5, the authors inadequately made a discrimination how to present the results: „3. Results […].“ Authors should arias these sentences because this confuses all readers.
These sentences were taken out.
Results
Overall, the result section appears too long and detailed; therefore, the results could perhaps be shortened a bit to underline the main developments. However, this is just a suggestion by the reviewer, if the authors do not agree, it might be ok to just slightly change it, or perhaps leave it in its current form.
While the results section of the article is divided into five parts, each analyzing the model from a distinct perspective, I have condensed the content of each section. This involves omitting extraneous details to enhance the overall readability and clarity of the text.
Discussion
The discussion is far too superficial. The authors should write a real discussion that integrates the current finding of the study into the bigger picture regarding the broad spectrum of literature of the last 5 to 10 years, for example. The current Discussion is scientifically insufficient! The authors should be guided by the literature (including the cited refs) suggested by the reviewer for the introduction. Despite stell cell medicine being a domain mainly for T1Dm because of its specific pathophysiology; Moreover, other novel stem cell approaches to cure particularly T2Dm should be mentioned and critically discussed. The authors should briefly summarize the main findings of the following papers and cite here: 1.) Zayzafoon M et al., J Cell Biochem 2000. doi: 10.1002/1097-4644(20001101). 2.) Yu S, et al., Stem Cell Res Ther 2019. doi:10.1186/s13287-019-1474-8. 3..) Bentz K et al., Cell Physiol Biochem 2010. doi:10.1159/000323991. 4.) Päth G et al. Metabolism 2019. doi:10.1016/j.metabol.2018.10.005. However, using these approaches requires caution, please discuss and cite: 1.) Molcanyi M, et al., J Neurosci Methods 2013. doi: 10.1016/j.jneumeth.2013.02.012. 2) Bora J et al., Naunyn Schmiedebergs Arch Pharmacol 2023. doi:10.1007/s00210-023-02631-1.
The discussion has been enriched (additional references were added).
Conclusions
There is a headline titled "5. Conclusions" however any text body is missing?! The author should choose either a conclusive sentence as the final one of the discussion or a full paragraph #5 that draws and summarizes the conclusions of this paper. But in its current form, the manuscript text does not make sense at the end of the discussion. The authors have to revise this part.
The Conclusions paragraph is added.
Overall, a critical or at best a native speaker should double-check the paper regarding typos and English grammar and spelling one on hand, and perhaps all authors should help to substantially improve the current manuscript version.
The paper has been double-checked regarding typos and English grammar and spelling.

Round 2
Reviewer 1 Report
The authors have addressed my concerns.
NA
Reviewer 2 Report
Please include a space between the Conclusion Paragraph and the supplementary material paragraph.
That‘s it. The Ms have been enormously improved, and appears to be ready for publication.
Kind regards
Your reviewer